# Characterisation of facial expressions and behaviours of horses in response to positive and negative emotional anticipation using network analysis

Romane Phelipon[1]*, Léa Bertrand[1], Plotine Jardat[1], Fabrice Reigner[2], Kate Lewis[3], Jérôme Micheletta[3], Léa Lansade[1]*

1 INRAE, CNRS, Université de Tours, PRC, Nouzilly, France, 2 UEPAO, INRAE, Nouzilly, France, 3 Department of Psychology, Centre for Comparative and Evolutionary Psychology, University of Portsmouth, Portsmouth, United Kingdom

* rphelipon.pro@gmail.com (RP); lea.lansade@inrae.fr (LL)

## Abstract

The welfare of an animal is closely linked to their emotional experiences, making it essential to identify reliable indicators of these emotions. This study aimed to identify behaviours and facial movements in horses experiencing contrasting emotional valence, triggered by the anticipation of a positive condition (going to pasture) or a negative condition (going alone to a novel environment). Twenty horses were daily trained to wait in a starting box before being exposed to these two conditions. After one week of positive training or negative training, we analysed horses' behaviours, cortisol variations, and facial movements while they waited in their starting box. First, we confirmed that the two conditions induced contrasting emotional valence, as evidenced by the shorter time taken to approach in the positive condition compared to the negative, and by the higher maximal heart rate and cortisol variation in the negative condition. Then using the Equine Facial Action Coding System (EquiFACS) and network analysis (NetFACS) we revealed distinct behaviours and facial expression profiles. In positive anticipation, the horses exhibited a greater range of behaviours, including shaking their heads from side to side, stepping back, sniffing, and pawing at the ground. Additionally, two distinct facial expression profiles were identified as specific to positive and negative anticipation. In positive anticipation, the horses displayed a higher neck, accompanied by a greater frequency of half-blinks and mouth movements. Conversely, in negative anticipation, the horses exhibited a medium neck, with ears backward accompanied by more flattened ears and expressed more nostril movements. The findings highlight the importance of these indicators in characterising horses' emotions and emphasise their significance for assessing equine welfare.

**Data availability statement:** All dataset files are available from the online repository: https://doi.org/10.57745/C9ND50 (INRAE data repository).

**Funding:** 'This work was funded by the EQUIACTION funds - https://equiaction.org/ (RP, LL) and IFCE: French Horse and Riding Institute - https://www.ifce.fr/ (RP, LL). These funding sources have no role in the study design, data collection and analyses or in the decision, preparation and submission of the manuscript'.

**Competing interests:** The authors have declared that no competing interests exist.

## Introduction

The well-being of animals is indissociably linked to their emotions, both positive and negative. Consequently, in order to assess well-being, it is also necessary to be able to assess emotions. To date, there is no single definition of emotion [1–3], however, in 2010 Mendl and his colleagues proposed a two-dimensional definition [4]. This approach defines an emotion in terms of its valence (positive or negative) and its arousal (low or high). For instance, fear would be characterised by a negative valence and high arousal, while relaxation would be characterised by a positive valence and low arousal.

A major challenge is to identify indicators of these emotional states. This can be achieved by directly observing animals in situations known to have intrinsically positive valence (e.g., providing palatable food) or negative valence (e.g., isolating a social animal). This approach involves analysing the immediate phase of emotion, which overlaps with what can be referred to as the 'consummatory' phase, as seen in studies seeking emotional indicators during play in dogs [5] or while being groomed manually or with automatic brushes in horses [6,7]. However, analysing indicators during this immediate or consummatory phase can be complicated, as the animal is often directly engaged in an action (such as eating or exerting effort) or may be difficult to observe (e.g., when transported in a trailer).

An alternative solution is to examine the anticipatory phase of these events (pre-consummatory phase). This has been done in several studies in different species [8–14]. To illustrate, a study of domestic fowl utilising Pavlovian conditioning revealed that the birds exhibited increased relaxation behaviour during the positive anticipation phase and conversely, displayed heightened locomotion behaviour during the negative anticipation phase [8]. If we can identify indicators of positive or negative emotional anticipation, it may also help determine whether the subsequent event is perceived positively or negatively, especially in cases where the intrinsic valence of these events is unclear. This is particularly relevant for horses, where the question arises as to whether they enjoy certain events they encounter during their life, such as participating in races or competitions. Identifying indicators of positive or negative event anticipation could provide valuable insights into the valence of these experiences.

Therefore, the question arises regarding the types of measures needed to find these emotional indicators. This can be achieved through various approaches, including physiological and behavioural indicators, or more recently, through the study of facial expressions [6,10,15–20], characterised by a combination of different facial movements.

Regarding the physiological variables, a classic indicator is the cortisol level, as short-term changes in blood or salivary cortisol levels often occur in response to an induced emotional state. Generally, it is reported to increase in response to stressful events in domesticated farm animals [21], including horses [22–25]. By contrast, some studies report a decrease in cortisol level in more positive situations such as during affiliative interactions in dogs [26] or during positive anticipation in horses [27]. Another well-established physiological indicator reflecting the arousal of a

situation is the heart rate [28,29]. In horses, numerous studies have demonstrated an increase in heart rate in the context of fear or stress [18,30–32]. Behaviour can also provide insights into an individual's emotional state. Studies have demonstrated an increase in activity during positive anticipation in rats [33], ferrets [9] pigs [34] and sheep [35]. The experience of negative anticipation has been demonstrated to induce stress in animals, resulting in behavioural responses such as increased time spent in a cage [9] or freezing behaviour in mice [36]. In Equids, studies have shown that positive anticipation contexts induce an overall increase in locomotor activity [11,37], a decrease in maintenance behaviour [11], and an increase in exploratory behaviour [27]. These behaviours, as well as avoidance and attraction behaviours, provide information about the valence of a stimulus or situation [4,38]. Finally, the analysis of facial movements can be employed as a promising indicator for characterising facial expressions linked to emotional states. A number of studies have described different facial expressions or specific facial movements such as in mice [15], foxes [39], cows [40] and horses [37,41–43], which has facilitated the identification of emotional states. For instance, in horses, recent studies have demonstrated that specific facial movements are indicative of negative emotional states, including food frustration [37,44] as well as social isolation and road transportation [17]. Furthermore, research indicated that facial movements in horses may also indicate positive emotional states, such as during gentle grooming [6] or feeding reinforcement [19,37]. A promising method to analyse facial movements across species [45] is the Facial Action Coding System (FACS). This method, initially developed for humans [46] and later adapted for various species, including horses [47], consists of the objective and accurate identification of the muscular movements (or action units, hereafter: AU) of the animal's face, enabling comparisons to be made under different conditions [48]. In addition to being an objective and standardised method, this approach is particularly relevant for horses, as they possess 17 primary facial action units, which is more than dogs, who have 16, or chimpanzees, who have 13 [47]. In horses, studies have been conducted using the EquiFACS system [47] in rather negative contexts, such as frustration [44,49] or fatigue [42]. To the best of our knowledge, only one study presents the use of this system in the positive context of positive anticipation, but no facial movements were specifically associated [44]. Two additional studies on positive food reinforcement demonstrate increased attention in horses, as evidenced by their ears being oriented forward more often, fewer eye blinks, and an increase in nose movements [19] and micro-movements of the mouth [50].

Moreover, a recent and novel statistical analysis method has been proposed to analyse data obtained using FACS: NetFACS [51]. Until now, FACS were analysed action unit by action unit (AU by AU), focusing on each facial movement independently. NetFACS allows for the calculation of the probability of co-occurrence (when action units are expressed together) between action units in a specific context. In addition to providing a more comprehensive and standardised method to analyse FACS data, NetFACS also provides more visual illustrations of facial movements in different contexts. Facial expressions are therefore conceptualized as a network of facial action units that interact with each other, offering a more accurate reflection of their complexity compared to traditional analyses. It has already been employed in the study of human facial movements [51], primate communication [52,53], and more recently, horse-to-horse interactions [48].

The objective of this study was to identify new indicators of emotional valence associated with positive and negative anticipation, based on facial expressions and behaviours. Two distinct procedures were employed to induce anticipatory responses in the individuals. One procedure involved the potential for access to a pasture, which was expected to be perceived as a positive stimulus, while the other involved social isolation in a box with predator sounds and new objects, which was expected to be perceived as a negative stimulus. First, we aimed to validate our two experimental procedures by demonstrating that they indeed induce contrasting emotional valences. To achieve this, we utilised established behavioural and physiological indicators, namely approach latency, heart rate and cortisol variation [2,4,38,54]. We hypothesized that horses would show lower cortisol variation in the positive condition (going to pasture) compared to the negative condition (social isolation) and would take less time to approach the pasture than to enter the social isolation box, reflecting attraction versus avoidance. Second, we aimed to identify new indicators of these two emotional states, based on specific behavioural responses and facial movements in horses. We hypothesized that our analysis, particularly

through network analysis using the NetFACS method [51], would reveal distinct behavioural profiles and facial expressions specific to each condition. These new indicators could serve as reliable indicators of emotional valence associated with positive and negative anticipation in the future.

## Materiel and methods

### Ethics statement

This study was approved by the Val de Loire Ethics Committee (CEEA VdL) and received a positive recommendation (authorisation number: CE19 - 2024-1302-1). Animal care and experimental treatments adhered to French and European regulations for housing and caring for animals used in scientific research (European Union Directive 2010/63/EU). All procedures were conducted under the authorisation and supervision of official veterinary authorities (agreement number F371752 delivered to the UEPAO animal facility by the veterinary service of the Indre-et-Loire department, France). Animals were not subjected to food deprivation during the experiment, and did not experience any invasive procedures. A 'limit point' was established before the start of the experiment to indicate when the distress of an experimental animal should be stopped. In the event that physical encouragement, such as tightening the lead rope or exerting pressure on the horse, was deemed necessary to facilitate movement, the procedure was halted. The individual was then returned to the living quarters, and horses were never tested again in this condition. In addition, for ethical reasons and because herd animals can experience stress when isolated, the horses were never left alone except during the social isolation phase in the negative condition (=2 minutes). Therefore, a familiar companion horse (not involved in the tests) was always visible to the tested horse during all experimental phases, except during the social isolation phase (under the negative condition).

### Subjects and groups

Twenty Welsh mares from the Experimental Unit UEPAO (INRAE Nouzilly 37380, France), aged between 4 and 11 years (7.3±2.6) were studied. They lived in groups in indoor stalls bedded with straw and had direct, unrestricted access to an outdoor paddock without grass. During the study, which took place between late February and early March, the horses did not have access to pasture, as they were not turned out to graze by the caretakers during this period. Hay and water were available ad libitum. These horses were used exclusively for research purposes and were handled daily by humans.

The twenty individuals were divided into two groups. During the first week of experimentation, ten individuals were trained and tested exclusively in the positive anticipation condition, while the remaining ten were tested in the negative anticipation condition. The following week, the groups were reversed, such that each individual was tested in both conditions.

### Experimental protocol

The procedure consisted of training the horses to undergo conditions of opposite valence. In the positive condition, the horses were trained to anticipate a positive event (access to a pasture), whereas in the negative condition, the horses were trained to anticipate a negative event (social isolation).

The experimental procedure lasted 12 days, consisting of a Pre-Training Day, a 3-day Training phase and a Test Day for each condition over two consecutive weeks, separated by a 2-day rest period (Fig 1).

The experimental set-up consisted of two different devices. The positive condition set-up consisted of an enclosed starting box with a door leading to a ten-metre corridor leading to a pasture (Fig 2A). The negative condition set-up consisted of an enclosed starting stall with a door leading to a ten-metre corridor leading to a stall unknown to the horses, containing new objects (plastic bags and a plastic tunnel) (Fig 2B).

**Pre-Training Day.** The Pre-Training Day corresponded to the first day of the experimental procedure for each condition. This consisted of placing the individual in a starting box, secured on both sides of the halter with lead ropes,

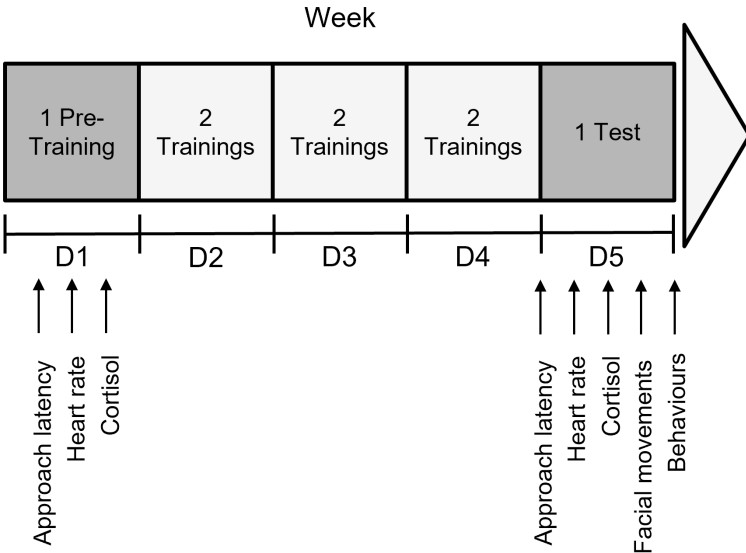

**Fig 1. Schematic representation of the experimental protocol conducted with 20 horses.** Each condition (positive and negative) included one Pre-Training session (Day 1), six training sessions (two per day on Day 2 to Day 4), and one Test session (Day 5). Measurements of approach latency, heart rate, cortisol levels (before and after the session) were taken during Pre-Training and Test sessions. Measurements of behaviours and facial movements were taken on Test sessions. The first group of horses (n = 10) experienced the positive condition during the first week, followed by the negative condition in the second week. Conversely, the second group of horses (n = 10) underwent the negative condition in the first week, followed by the positive condition in the second week.

and facing a closed door. The anticipation phase was marked by the opening of the door by an experimenter standing out of sight, giving the individual a view of a ten metres long corridor. After 30 seconds, the door was closed again and opened a second time for a further 30 seconds. At the end of the thirty seconds, the experimenter untethered the lead ropes from the horse's halter to allow the subject access to the ten metres corridor.

In the positive anticipation condition, the corridor led to a pasture (Fig 2A). The pasture is known to have a positive impact on horses [55,56] and represented a familiar, highly valued environment, particularly as the individuals did not have access to it during the test period. To strengthen the association of the situation with a permanent cue, the experimenter opening the door was always dressed in a green coat. When the individual was untied after the two thirty-second periods marking the positive anticipation phase, she was free to move around the experimental set-up and access the pasture. If the individual was still in the starting box or in the corridor after the thirty seconds allowed, the experimenter guided her with a lead rope towards the pasture. After a period of two minutes in the pasture, the individual was given three handfuls of palatable concentrate feed and then returned to the living quarters.

In the negative anticipation condition, the corridor led to a box that was initially unfamiliar to the individuals (social isolation box) containing new objects (Fig 2B). To strengthen the association of the situation with a permanent cue, the experimenter opening the door was always dressed in a blue coat. When the individual was untied after the two thirty-second periods marking the negative anticipation phase, she was free to move around the experimental set-up and access the social isolation box. If the individual was still in the starting box or in the corridor after thirty seconds allowed, the experimenter guided her with a lead rope to the isolation box. The individual then underwent a two-minute period of social isolation, without visual contact with conspecifics, in a box containing new objects (five plastic bags hanging from the walls and one plastic tunnel, which changed position from day to day) and a loudspeaker broadcasting recordings of predator calls [57]. The sound level ranged between 60 and 75 decibels, depending on the horse's position, as she was free to move

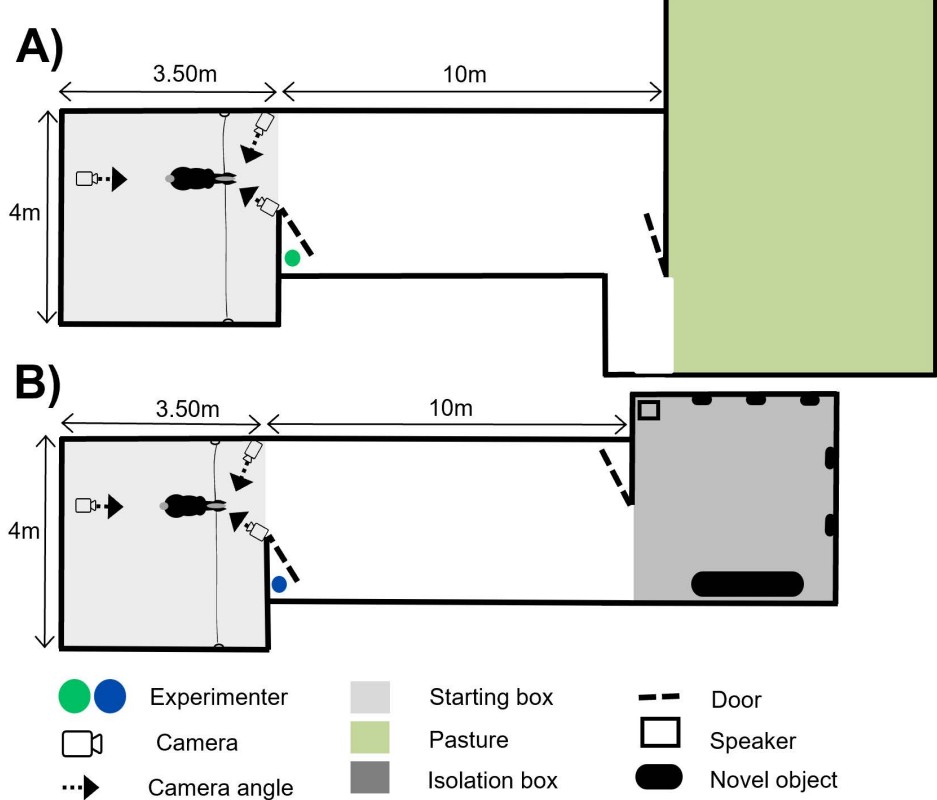

**Fig 2. Schematic representation of the experimental set-up.** 20 horses were trained and tested in the A) positive anticipation set up B) negative anticipation set up.

around in the isolation box. Once the two minutes of social isolation had elapsed, the procedure was terminated, and the individual was returned to the living quarters.

**Training.** The Training Days were conducted in accordance with the established procedure of the Pre-Training Day, with the exception of two adaptations. Firstly, the experimenter did not open the door a second time during the anticipation phase. Secondly, once the horses had undergone the positive condition (being in the pasture for two minutes) or the negative condition (being in social isolation for 2 minutes), they were not returned directly to their living quarters but the entire procedure was carried out a second time immediately.

**Test Day.** The Test Day corresponded to the last day of the procedure, after the training period. It followed exactly the same procedure as the Pre-Training Day.

### Number of sessions

We planned for each horse to undergo 1 Pre-Training, 6 training sessions (2 per day), and 1 Test in each condition (Fig 1). Prior to the experiment, we also defined a 'limit point' with the regional ethics committee for animal experimentation, which indicated the point at which the distress of an experimental animal had to be stopped (see the ethics statement section of the experimental protocol above). This limit point was reached when one horse did not spontaneously follow the experimenter, and it became necessary to physically encourage the horse to move forward by tightening the lead rope or pushing in any way. After 5 sessions of training in the negative condition, this limit was reached for one individual.

Consequently, we stopped the procedure for all animals, as previously decided, and all the animals were subjected to 5 sessions of the negative condition instead of 6. For the positive anticipation condition, we conducted the initially planned number of sessions. The difference in the number of training sessions was not an issue with regard to the goal of this study, which was to induce, by the end of the procedure, emotional reactions contrasting in valence, regardless of the number of sessions required to achieve this.

**Data collection**

**Approach latency.** The time taken for the horses to enter the pasture or the social isolation box spontaneously (without being led by the experimenter) was recorded on the Pre-Training Day and the Test Day for each condition (Fig 1). This time was measured from the moment the horse was untied by the experimenter until the moment the horse placed their fourth foot in the pasture or the isolation box.

**Heart rate.** For each horse, heart rate was monitored during Pre-Training and Test Days for each condition (positive or negative) using the Polar system recording (Polar Equine RS800CX Science, Polar Oy, Finland) on the tested horse (Fig 1). Horses were clipped in advance, in the areas where the external sensors would be positioned. During fitting, the external sensors, consisting of two electrodes placed on the withers and behind the left front leg, were applied with ultrasound gel to obtain a better signal. The recording started at the opening of the stall door, marking the anticipation phase, and stopped at the end of the anticipation phase, just before the test horse was released. After data collection, heart rate data was visually corrected in order to remove artifacts that might result from poor electrode-skin conductance or equipment malfunction and were detected visually, with the criterion being any isolated point creating a peak too rapid to be physiological in the heart rate curve [58]. Then the mean heart rate and the maximal heart rate were calculated. Out of 160 cardiac data recordings (40 representing the mean heart rates of each horse at Pre-Training Day, 40 at Test Day, and 40 representing the maximal heart rate at Pre-Training Day and 40 at Test Day), 26 recordings were excluded from analysis due to either equipment failure during data acquisition or insufficient data quality following artifact correction.

**Cortisol.** During Pre-Training and Test Days for each condition (positive or negative), two saliva samples were taken from each individual tested with Salivette® Cortisol (Sarstedt, Nümbrecht, Germany) to measure cortisol concentrations: one just before the anticipation phase and one fifteen minutes after. A cotton swab held by a metal clamp was kept in the horse's mouth between 30 and 40 seconds, positioned above and below the tongue before being placed in Salivettes. In order to avoid any effect of the daily cortisol cycle, each horse was always tested at the same time of the day. Once collected, the saliva samples were centrifuged for ten minutes (Heraeus Megafuge 40R, 4°C, 2500g), then supernatant was placed in 1mL micro-tubes before being frozen at -20°C. The analyses were conducted at the INRAE Laboratory in Nouzilly, France. The concentration of cortisol in saliva was determined using enzyme immunoassay (EIA), which involved a competition between the cortisol in the saliva and a cortisol alkaline phosphatase solution. Cortisol was measured in two 50 μL aliquots, and all samples were analysed in duplicate within the same assay. The limit of detection of the assay was 2 ng/mL. The mean value of cortisol concentration from the two 50 μL aliquots was retained. At the end of the study, eight samples had been collected from each individual (Fig 1). The variation in salivary cortisol was calculated for each horse in each condition as the variation between the mean value of cortisol of the saliva sample taken after the anticipation phase minus the mean value of cortisol of the saliva sample taken before the anticipation phase. The total cortisol variation was then calculated as the variation in cortisol on the Test Day minus the variation on the Pre-Training Day.

**Behaviours and facial movements.** Behavioural and facial movements data were collected for each horse on the Test Day for each condition (positive or negative) (Fig 1). A camera (Sony FDR-AX53 4K) was positioned so as to have a wide angle of view, making it possible to film the behaviour or positions of certain parts of the individual's body during the anticipation phase. Two other cameras were fixed to the wall on either side of the tested subject and oriented towards the head, providing an optimum view for video analysis of facial movements. These cameras recorded continuously during the anticipation phase. A total of 80 videos were recorded and analysed, with four videos per subject: one capturing

behaviours in the positive condition, one focusing on facial movements in the positive condition, one capturing behaviours in the negative condition, and one focusing on facial movements in the negative condition. These videos were coded using BORIS software [59]. After excluding periods when the individual was outside the camera's field of view, the analysable anticipation time was used for video coding, including behavioural data (mean±SD: 35.19±3.04 seconds) and facial movements data (mean±SD: 30.97±6.04 seconds).

For facial movements, the eyes, the lower part of the face (nostrils and lips) and part of the ears were coded according to the action units (AU) and action descriptors (AD) of EquiFACS, a system for coding facial actions in domestic horses [47]. One certified EquiFACS coder coded all the videos. To ensure the reliability of the coding of behaviours and facial movements, a second certified EquiFACS coder independently coded 10% of the videos, which were randomly selected. This second coder was blind to the condition, as the starting boxes were similar and the angle of the camera did not show the pasture or the isolation box. The same proportion of videos was coded in positive and negative conditions.

All the action units and descriptors listed for the horse (Table 1) were used, following the recommendations of the Equi-FACS. For practical reasons, the ears action descriptor EAD101 'Ears Forward' and EAD104 'Ear Rotator' were coded in one-second scan intervals as realised in other studies in horses [6,37]. Also, EAD103 'Ear Flattener' was coded in duration and not in occurrence. For simplicity of reading, action units (AU), action descriptors (AD), Ear action descriptors (EAD) and position of the ears and neck will be referred to as 'action units' in the remainder of this text.

The behaviours and body positions observed were previously defined by an ethogram based on bibliographical knowledge [11,44] of horse behaviour in positive and negative contexts (Table 1). All the behaviours were coded continuously over the duration of the video (all occurrences method [60]), in terms of duration or occurrence. The position of the neck was characterised based on the angle between the line from the withers to the ears and the horizontal line: if it was less than 0 degrees the neck was defined as low, if it was between 0 and 30 degrees the neck was defined as medium, and if it was greater than 30 degrees the neck was defined as high.

## Statistical analysis

All statistical analyses were carried out using the version 4.4.0 of RStudio software [61], with the significance threshold set at 0.05.

**Inter-coder and Intra-coder reliability.** The level of agreement on the coded action units was calculated using two-factor intra-class correlation coefficients (ICC) for each action unit and each behaviour separately. Only the variables exhibiting at least 'moderate' agreement (>50) [62] were retained for subsequent data analysis. As the position of the ears was coded using a distinct methodology (categorial data), a Cohen's Kappa test was conducted. The same approach was employed for intra-coder reliability, whereby 10% of the selected videos were re-coded by the initial coder.

**Evolution and comparison of the behavioural and physiological indicators across conditions.** To assess the evolution of the behavioural and physiological indicators between the Pre-Training and Test Days in the positive and the negative condition, as well as their comparison on the Test Day, approach latency, mean heart rate, maximal heart rate, and cortisol variation were analysed using generalized linear mixed models (GLMM) with the *glmmTMB* package [61].

To analyse the evolution of the behavioural and physiological indicators between the Pre-Training and Test Days in the positive and negative conditions, the Day (Pre-Training Day or Test Day) was set as the fixed variable. Similarly, to compare these indicators on the Test Day between the positive and negative conditions, the variable condition (positive or negative) was used as the fixed variable. Two variables were incorporated into the explicative models as random effects: the individual, which accounted for inter-individual differences, and the order of conditions (positive then negative or negative then positive). The explicative model was then compared to a null model using an analysis of variance (ANOVA), to test whether the condition had a significant effect on the behavioural or physiological indicator considered. Then, the reliability of the model selected was evaluated using the *DHARMa* package [63] to check distributions, homoscedasticity

**Table 1. Variables used for video coding of 20 horses, derived from EquiFACS: The Equine Facial Action Coding System.**

| Category | Code | Variable name | Description | Coding type |
|---|---|---|---|---|
| Lower Face Action Units | AU113 | **Sharp Lip Puller** | Derived from the EquiFACS method, see [47]. | Occurrence |
| | AU16 | **[Lower Lip Depressor]** | | Occurrence |
| | AU17 | **Chin Raiser** | | Occurrence |
| | AU10 | **Upper Lip Raiser** | | Occurrence |
| | AU12 | Lip Corner Puller | | Occurrence |
| | AU18 | Lip Pucker | | Occurrence |
| | AU122 | Upper Lip Curl | | Occurrence |
| | AU25 | Lips Part | | Occurrence |
| | AU26 | Jaw Drop | | Occurrence |
| | AU27 | Mouth Stretch | | Occurrence |
| | AUH13 | **Nostril Lift** | | Occurrence |
| | AU24 | Lip Presser | | Occurrence |
| Lower Face Action Descriptors | AD160 | Lower Lip Relax | | Occurrence |
| Ear Action Descriptor | EAD103 | **Ear Flattener** | | Duration |
| | EAD101 | (Ears Forward) | | Duration |
| | EAD102 | Ear Adductor | | Duration |
| | EAD104 | (Ear Rotator) | | Duration |
| Upper Face Action Units | AU145 | **Blink** | | Occurrence |
| | AU47 | **Half Blink** | | Occurrence |
| | AD1 | **Eye White Increase** | | Occurrence |
| | AU101 | **Inner Brow Raiser** | | Occurrence |
| | AU5 | **Upper Lid Raiser** | | Occurrence |
| | AU143 | Eye Closure | | Occurrence |
| Miscellaneous Actions | AD19 | **Tongue Show** | | Occurrence |
| and Supplementary Codes | AD29 | Jaw Thrust | | Occurrence |
| | AD30 | Jaw Sideways | | Occurrence |
| Gross Behaviours Codes | AD38 | **Nostril Dilator** | | Occurrence |
| | AD133 | [Blow] | | Occurrence |
| | AD81 | **Chewing** | | Duration |
| | AD50 | Vocalization | | Occurrence |
| | AD84 | **Head Shake Side to Side** | | Occurrence |
| | AD85 | Head Nod Up and Down | | Occurrence |
| | AD80 | Swallow | | Occurrence |
| Supplementary Ears Position | – | *Ears forward* | Derived from the one-second scan intervals, see [6,37]. | Occurrence |
| | – | *Ears backward* | | Occurrence |
| | – | *On the side ears* | | Occurrence |
| | – | *Asymmetrical ears* | | Occurrence |
| Supplementary Neck Position | – | *High Neck Position* | Angle from the withers to the ears >30°. | Duration |
| | – | *Medium Neck Position* | Angle from the withers to the ears between 0° and 30°. | Duration |
| | – | *Low Neck Position* | Angle from the withers to the ears <0°. | Duration |
| Supplementary Behaviours | – | *Observe Conspecific/Experimenter* | Stares at the experimenter or the accompanying horse. | Duration |
| | – | *Step Back* | Moves at least two limbs backwards. | Occurrence |
| | – | *Step Further* | Moves at least two limbs forward. | Occurrence |
| | – | *Sniff the Ground* | The head is lowered towards the ground and the nostrils dilate. | Duration |
| | – | *Paw the Ground* | The hoof rises up and scratches the ground. | Occurrence |

Variables for which the strength of inter-reliability was less than 'moderate' are shown in brackets and uncoded variables are in parentheses. Variables in italics are additional variables not included in the EquiFACS added in the present study. In bold: variables for which the coding prevalence in this study was over 10%.

of the residuals and within-group variance. For explicative models that explained data variability significantly better than the null model, an ANOVA was carried out using the *Anova ()* function from the *car* package [64].

In the negative condition, all horses consistently took more than 30 seconds for the approach latency variable during both the Pre-Training and Test Days, resulting in no variability within the data. Because the data were censored at the maximum value of 30 seconds, there was a total lack of within-group variation. This made it impossible to perform statistical analyses using generalized linear mixed models (GLMM), as the absence of variation prevented accurate estimation of model parameters.

**Characterisation of specific behaviours and facial expressions in response to positive and negative conditions.** Prior to analysis, behavioural and action unit data (Table 1) were transformed into frequencies to account for the unequal time periods spent by each individual within the camera's field of vision.

$$Occurence\ frequency = \frac{number\ of\ occurences\ \times\ 100}{visible\ time\ (sec)}$$

$$Duration\ frequency = \frac{duration\ of\ behaviour\ \times\ 100}{visible\ time\ (sec)}$$

Each behaviour and action unit was investigated using the same method as described above using generalized linear mixed models. The explicative model took as a fixed variable the type of condition (positive or negative) and as random effects the individual, which accounts for inter-individual differences, and the order of conditions (positive then negative or negative then positive).

**NetFACS analysis.** The NetFACS package for R [51] was used to explore interconnections between simultaneously expressed action units, which enables network-based analysis of action units rather than examining them in isolation. Details of the methodology are provided in Mielke et al. [51] and in the NetFACS manual (https://github.com/NetFACS).

To achieve this, the BORIS software [59] was employed to export the dataset of all the negative and positive anticipation phases of each horse in the form of a binary matrix. For each action unit, the matrix indicated the presence (1) or absence (0) of the unit at each second. In this manner, the different action units expressed during a scan constituted a combination of action units.

To represent the combinations of action units between them (their co-occurrences), in order to obtain the visual representation of the positive and negative anticipation face, we used the conditional probability network. This tool highlights the significant connections between the action units expressed simultaneously, providing a clear view of the interactions between them. Only action units for which the coding prevalence in the study was over 10% were included [44] (Table 1). The 'Medium Neck Position 'was considered the baseline position; therefore, the two selected neck positions were the 'High Neck Position' and the 'Low Neck Position', as they provide more detailed information about horse's arousal state and were considered as events.

## Results

### Inter-coder and Intra-coder reliability

Following the second coding by a different EquiFACS certified coder, three of all the action units and behaviours analysed had a level of concordance that was insufficient for inclusion in the statistical analyses (S1 Table, 'poor' level agreement). Thus, 'Blow' (AD133), 'Lower Lip Depressor' (AU16) and 'Head Nod Up and Down' (AD85) were not analysed (Table 1). All variables showed at least a 'moderate' level of agreement for intra-encoder reliability (S2 Table).

### Evolution and comparison of the behavioural and physiological indicators across conditions

**Evolution of behavioural and physiological indicators between the Pre-Training Day and the Test Day in the positive and negative conditions.** For the positive condition, a decrease was observed on the Test Day compared

to the Pre-Training Day for the approach latency (GLMM, p-value <0.001, Table 2, Fig 3), the mean heart rate (GLMM, p-value = 0.001, Table 2, Fig 3), the maximal heart rate (GLMM, p-value = 0.03, Table 2, Fig 3), and the cortisol variation (GLMM, p-value = 0.006, Table 2, Fig 3).

In the negative condition, no statistically significant difference was observed in the approach latency between the Pre-Training Day and the Test Day (Table 2). The mean heart rate decreased on the Test Day compared to the Pre-Training Day (GLMM, p-value = 0.008, Table 2, Fig 3), but not the maximal heart rate (GLMM, p-value = 0.87, Table 2, Fig 3). The cortisol variation tended to increase on the Test Day compared to the Pre-Training Day (GLMM, p-value = 0.06, Table 2, Fig 3).

**Comparison of physiological indicators between the positive and negative conditions on the Test Day.** The maximal heart rate was higher in the negative condition compared to the positive condition on the Test Day (GLMM, p-value = 0.02, Table 3, Fig 3). There was no significant difference in the mean heart rate between the positive and negative conditions on the Test Day (GLMM, p-value = 0.12, Table 3). The total cortisol variation was significantly higher in the negative condition than in the positive condition on the Test Day (GLMM, p-value = 0.001, Table 3, Fig 3) and the cortisol variation tended to be higher in the negative compared to the positive condition (GLMM, p-value = 0.09, Table 3).

### Characterisation of specific behaviours and facial expressions in response to positive and negative conditions

With regard to the behaviours, AD84 'Head Shake Side to Side' (GLMM, p = 0.001), as well as the 'Step Back' (GLMM, p-value = 0.02), 'Sniff the Ground' (GLMM, p-value <0.001) and 'Paw the Ground' (GLMM, p-value = 0.03) were exhibited with greater frequency by the horses in the positive anticipation compared to the negative anticipation on the Test Day (Table 4). In addition, results showed that horses in positive anticipation expressed more 'High Neck Position' than in negative anticipation (GLMM, p-value = 0.02, Table 4). Conversely, in negative anticipation, horses expressed more 'Medium Neck Position' than in positive anticipation (GLMM, p-value = 0.03, Table 4).

For the facial movements, the results showed that the action units AU113 'Sharp Lip Puller' (GLMM, p-value = 0.003) and AU47 'Half Blink' (GLMM, p-value <0.001) were expressed more by horses in positive anticipation than in negative anticipation whereas EAD103 'Ear Flattener' (GLMM, p-value <0.007) was expressed more in negative anticipation than in positive anticipation on Test Day (Table 4).

### NetFACS

**Conditional probability networks.** Conditional probability networks represented the probability of co-occurrence of the action units in the positive and the negative anticipation phases (Fig 4).

**Table 2. Results of the comparison of approach latency (in seconds), heart rate (in bpm) and cortisol variation (in ng/mL) between the Pre-Training Day and the Test Day, using generalized linear mixed models on 20 horses.**

| Evolution of physiological indicators between the Pre-Training Day and the Test Day | | | | |
|---|---|---|---|---|
| Variable | Estimate | S.E. | Z | P-value |
| Approach latency in positive condition | -10.55 | 1.40 | -7.53 | **<0.001** |
| Approach latency in negative condition | x | x | x | x |
| Mean heart rate in positive condition | -20.82 | 6.63 | -3.13 | **0.001** |
| Mean heart rate in negative condition | -8.85 | 3.35 | -2.63 | **0.008** |
| Maximal heart rate in positive condition | -25.14 | 12.15 | -2.06 | **0.03** |
| Maximal heart rate in negative condition | – | – | – | 0.87 |
| Cortisol variation in positive condition | -1.74 | 0.64 | -2.70 | **0.006** |
| Cortisol variation in negative condition | – | – | – | 0.06 |

Bold values indicate those with a p-value <0.5. "x": analysis not possible due to absence of variability in the data. "-": no significant difference was found between the explicative and the null model.

## A) Evolution of the indicators between the Pre-Training Day and the Test Day

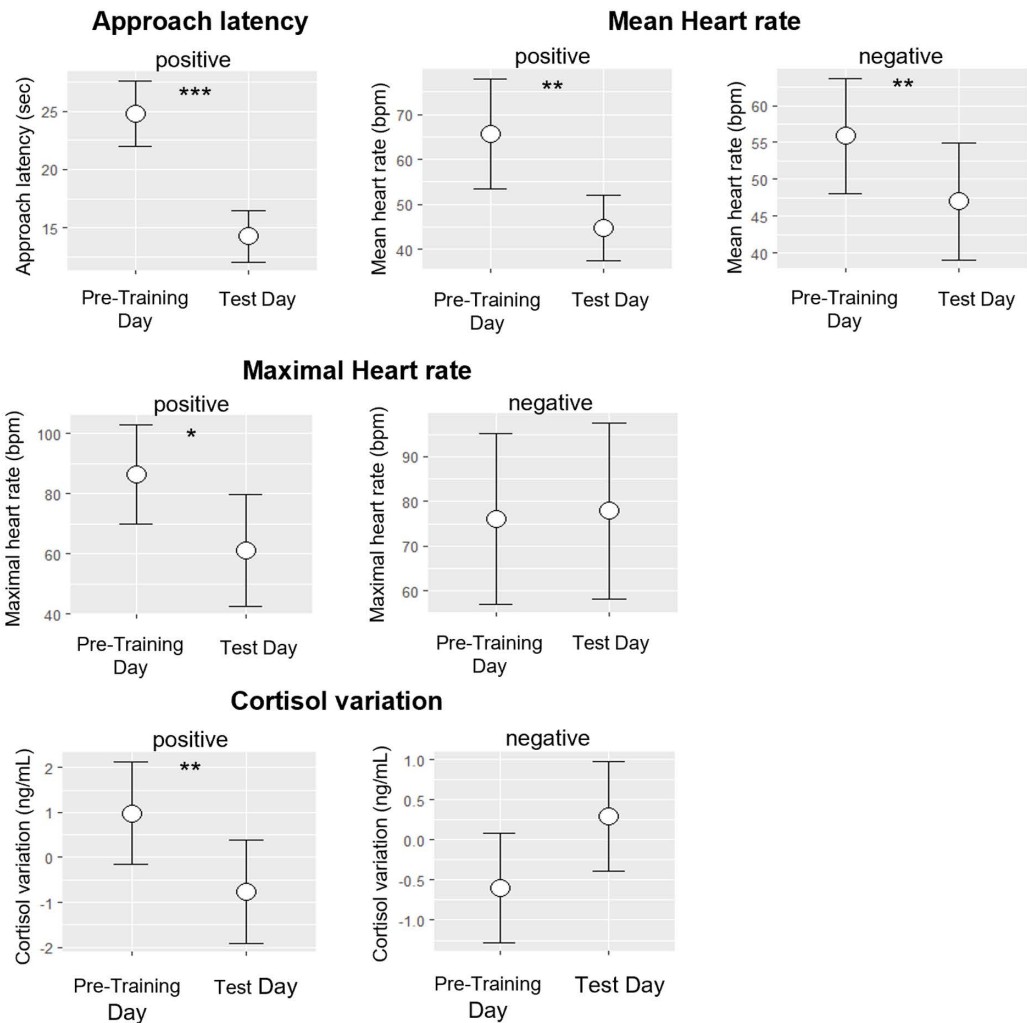

## B) Comparison of the indicators between conditions on the Test Day

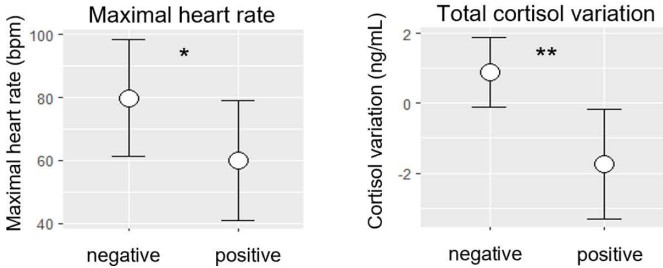

**Fig 3. Evolution and comparison of behavioural and physiological indicators during the procedure in 20 horses.** Means and standard errors from the corresponding models presented in Tables 2 and 3. **A)** Evolution of the indicators between Pre-Training Day and Test Day: approach latency, mean heart rate, maximal heart rate and cortisol variation in the positive and the negative condition. **B)** Comparison of the indicators between conditions on the Test Day: maximal heart rate and total cortisol variation rate between the positive and negative condition. *p-value < 0.05, **p-value < 0.01, ***p-value < 0.001.

**Table 3.** Results of the comparison of approach latency (in seconds), heart rate (in bpm) and cortisol variation (in ng/mL) between the positive and negative conditions on the Test Day, using generalized linear mixed models on 20 horses.

**Comparison of physiological indicators between conditions on the Test Day**

| Variable | Estimate | S.E. | Z | P-value |
|---|---|---|---|---|
| Approach latency | x | x | x | x |
| Mean heart rate | – | – | – | 0.12 |
| Maximal heart rate | -19.86 | 8.56 | -2.32 | **0.02** |
| Cortisol variation | – | – | – | 0.09 |
| Total cortisol variation | -2.63 | 0.82 | -3.18 | **0.001** |

Bold values indicate those with a p-value < 0.5. "x": analysis not possible due to absence of variability in the data. "-": no significant difference was found between the explicative and the null model.

Firstly, we observed that the positive and negative anticipation faces had the same base, composed of 'On the side ears', AU145 'Blink' and AU47 'Half Blink' connected only to AU101 'Inner Brow Raiser'.

Between the two types of anticipation, we also found AU5 'Upper Lid Raiser' and AD1 'Eye White Increase', with the difference that in positive anticipation they were connected to High Neck Position whereas in negative they were only connected to AU101 'Inner Brow Raiser'.

AD81 'Chewing' was also common to both of the two networks but was not connected in the same way: in positive anticipation it was connected to AD19 'Tongue Show' which was itself connected to 'Ears forward', whereas in negative anticipation it was connected to 'Ears backward'.

Finally, 'Ears forward' was also in both types of anticipation but was connected in a different way. In positive anticipation, this ear position was linked as mentioned above to AD19 'Tongue Show' and AD81 'Chewing', whereas in negative anticipation 'Ears forward' was linked to 'High Neck' position and AD38 'Nostril Dilator', itself linked to AUH13 'Nostril Lift'. There were therefore many action units that were common to both types of anticipation, but their combinations were very different.

**Table 4.** Results of the comparison between positive and negative condition for each facial action unit variable and behaviour, using generalized linear mixed models on the Test Day on 20 horses.

| Variable name | Estimate | S.E. | Z | P-value |
|---|---|---|---|---|
| Sharp Lip Puller (AU113) | 3.54 | 1.21 | 2.93 | **0.003** |
| Ear Flattener (EAD103) | -1.81 | 0.67 | -2.70 | **0.007** |
| Half Blink (AU47) | 10.08 | 2.25 | 4.50 | **<0.001** |
| Tongue Show (AD19) | 1.51 | 0.79 | 1.91 | 0.06 |
| Head Shake Side to Side (AD84) | 0.94 | 0.29 | 3.25 | **0.001** |
| Ears forward | -0.07 | 0.07 | -1.03 | 0.3 |
| High Neck Position | 0.66 | 0.28 | 2.40 | **0.02** |
| Medium Neck Position | -0.56 | 0.26 | -2.12 | **0.03** |
| Step Back | 0.80 | 0.33 | 2.42 | **0.02** |
| Sniff the Ground | 1.20 | 0.36 | 3.36 | **<0.001** |
| Paw the Ground | 0.78 | 0.37 | 2.12 | **0.03** |

Only results where the explicative model explained data variability significantly better than the null model are presented. Bold values indicate those with a p-value < 0.5.

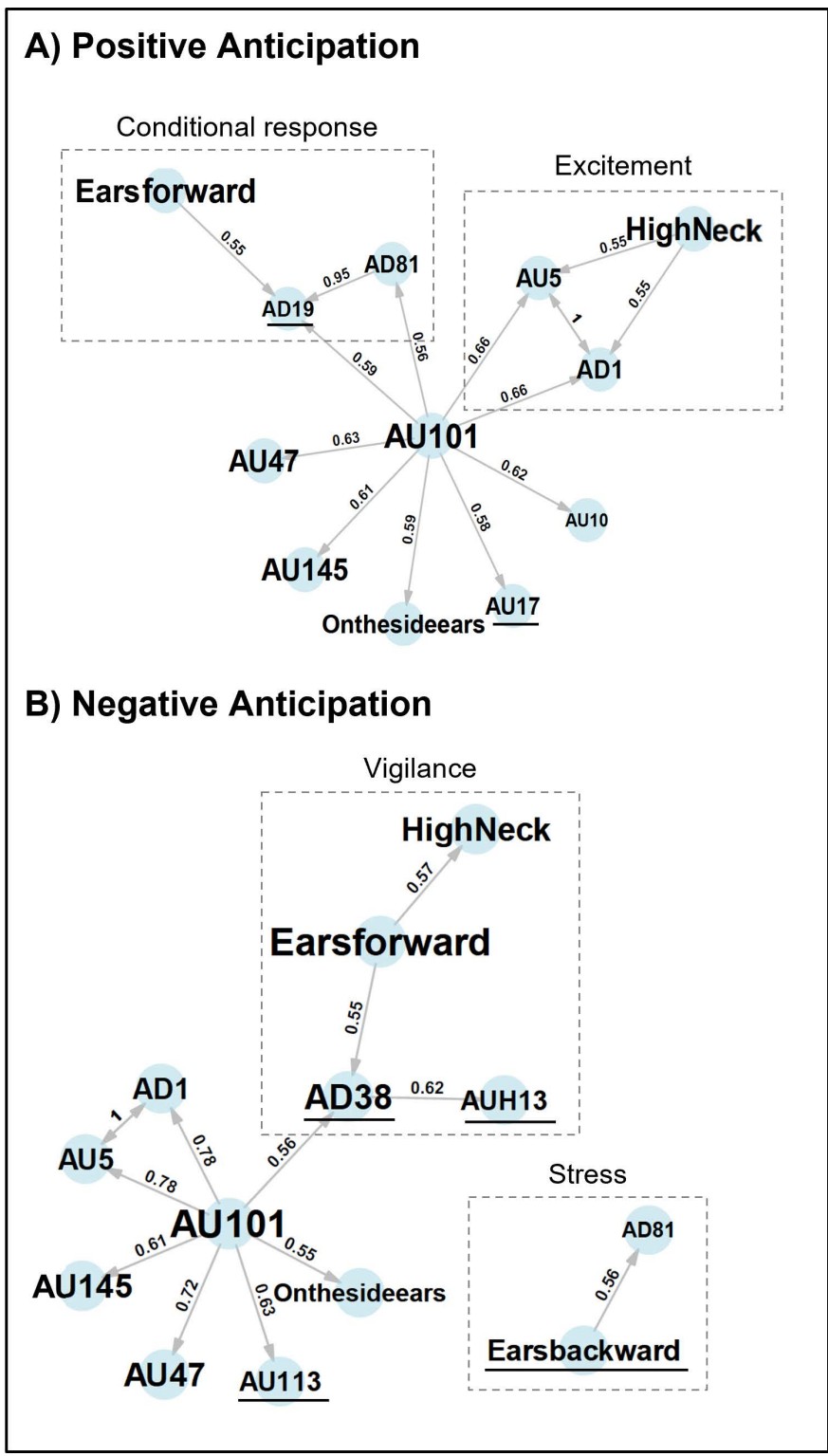

**Fig 4. Networks based on the conditional probabilities of the co-occurrence of actions units in 20 horses, in A) positive anticipation and B) negative anticipation.** Only action units with at least five co-occurrences, a significant association (p-value < 0.05), and probabilities exceeding 0.55 are displayed to facilitate understanding. The direction of activation of the connections is represented by the arrows linking the action units (i.e., AU47

'Half Blink' has a 63% probability of activating AU101 'Inner Brow Raiser' in the positive anticipation face). More frequent elements appear larger on the graph. Action units that are not common to both networks are underlined in black. The dotted boxes represent combinations of action units that can be interpreted as being specific to postures or behaviours.

Ultimately, a number of action units were identified as exclusive to a single type of anticipation. This was the case for AD19'Tongue Show', AU10 'Upper Lip Raiser' and AU17 'Chin Raiser' in positive anticipation and AD38 'Nostril Dilator', AUH13 'Nostril Lift', AU113 'Sharp Lip Puller' and 'Ears backward' in negative anticipation.

By combining the results of our two analyses, the individual analysis and the network analysis (Table 5), we propose two different facial expression profiles related to positive and negative anticipation (Fig 5).

## Discussion

The objective of this study was to identify the behavioural and facial movements responses of horses to conditions of opposite emotional valences: positive and negative anticipation. After validating the emotional valence of the two conditions using existing indicators, we identified two distinct profiles of behaviour and facial expression, each corresponding to one of the conditions.

### Evolution and comparison of the behavioural and physiological indicators across conditions

Firstly, we validated the emotional valence of the two conditions of the procedure, using existing indicators: approach latency, heart rate and cortisol level [21,28,38].

The horses took less time to reach the pasture (positive condition) compared to the isolation box (negative condition), where all horses exceeded the maximum time limit of 30 seconds on both the Pre-Training Day and the Test Day. Furthermore, in the positive condition, horses reached the pasture more quickly on the Test Day compared to the Pre-Training Day. Thus, horses, when given the choice whether or not to approach the pasture or the social isolation box, exhibited a clear preference for approaching the pasture over the social isolation box, which was avoided during all the procedure.

With regard to the physiological variables, in both conditions, the horses' mean heart rate decreased during the Test Day compared to the Pre-Training Day. Arguably, despite the contrasting circumstances, the horses became habituated to the situations (presence in the starting box). However, the maximal heart rate only decreased in the positive condition and not in the negative condition, indicating a higher arousal state. Additionally, on the Test Day, the horses exhibited a higher maximal heart rate in the negative condition compared to the positive condition. Thus, the positive and negative conditions appeared to have contrasting arousal states, with the negative condition having a higher intensity. Furthermore, this result suggests that horses experienced a negative valence in the negative condition, supported by previous reports in equids showing increased heart rate in response to fear or stress stimuli [32,65–67].

Finally, the variation in cortisol levels was lower in the positive condition on the Test Day compared to the Pre-Training Day, indicating that the horses were less stressed following the completion of the procedure. Conversely, in the negative condition, there was a tendency for cortisol variation to increase on the Test Day, suggesting that the horses continued to experience stress, or potentially heightened stress levels, after they underwent the negative condition procedure. Furthermore, the overall rate of change in cortisol remained higher in the negative condition than in the positive condition, indicating a more pronounced change in cortisol levels. Consequently, it can be deduced that the two conditions were contrasted in terms of valence, supported by higher cortisol levels in response to negative valence situations observed in horses [67–69], cows [70], sheep [71], dogs [72] and many other species [73].

The differences in these three indicators, well established in the scientific literature as measures of valence—namely, approach and avoidance behavioural responses, heart rate, and variations in cortisol levels—clearly validated the

**Table 5. Summary of the action units in 20 horses found to be related to positive anticipation ('P') and negative anticipation ('N') according to the different statistical analyses.**

| Variables | Glmm (Table 4) | Conditional probability networks (Fig 4) |
|---|---|---|
| AU113 'Sharp Lip Puller' | P | N |
| AU17 'Chin Raiser' | | P |
| AU10 'Upper Lip Raiser' | | P |
| AUH13 'Nostril Lift' | | N |
| EAD103 'Ear Flattener' | N | |
| AU47 'Half Blink' | P | |
| AD19 'Tongue Show' | | P |
| AD38 'Nostril Dilator' | | N |
| High Neck Position | P | |
| Medium Neck Position | N | |
| Ears backward | | N |

For the conditional probability networks, only elements identified as exclusive to a type of anticipation were noted as being related to positive anticipation ('P') and negative anticipation ('N').

contrasting valences of the two conditions: anticipation of a positive event (pasture) and anticipation of a negative event (isolation and novelty).

### Characterisation of specific behaviours and facial expressions in response to positive and negative anticipations

In a first step, we analysed single actions units and behaviours using generalized linear mixed models. This enabled us to determine which action units and behaviours were more often expressed in one type of anticipation compared with the other. The behaviours of pawing the ground, sniffing the ground, stepping back and shaking the head side to side were more frequently expressed in positive anticipation. These behaviours, alongside greater motor activity, have already been identified in a context of positive anticipation in horses [11,37], however these behaviours have also been observed as indicative of stress [74–76]. This prompts the question of whether these behaviours reflect the arousal experienced by the horse rather than its valence. To achieve a more precise understanding, it is therefore necessary to take into account supplementary elements, such as the analysis of facial movements.

The initial analysis revealed that in positive anticipation, the horses exhibited more high neck position and a greater prevalence of half blinks. Conversely, in negative anticipation, the horses displayed a more medium neck position and a higher incidence of ears flattening movements.

The position of the neck may vary depending on the emotional context. Some studies reported a less lowered neck during positive reinforcement [19], while others reported a high neck in negative or frustrating situations [6,37]. The divergence in findings indicates that neck position may be a more accurate reflection of the intensity of the situation than its emotional valence. However, the position of the neck can also be influenced by what the horse is observing, as horses need to raise their heads to focus, depending on whether it is near or distant, or located above or below their line of sight [77].

Conversely, the position of the ears is frequently regarded as an indicator of valence [78,79]. Some studies showed that backward ears were observed more often in negative contexts [37,41,78] in agreement with our results which showed a more frequent expression of flattened ears in negative anticipation, however another one observed backward ears in a positive context [6]. These divergent results underline the importance of analysing action units in combination with each other rather than in isolation.

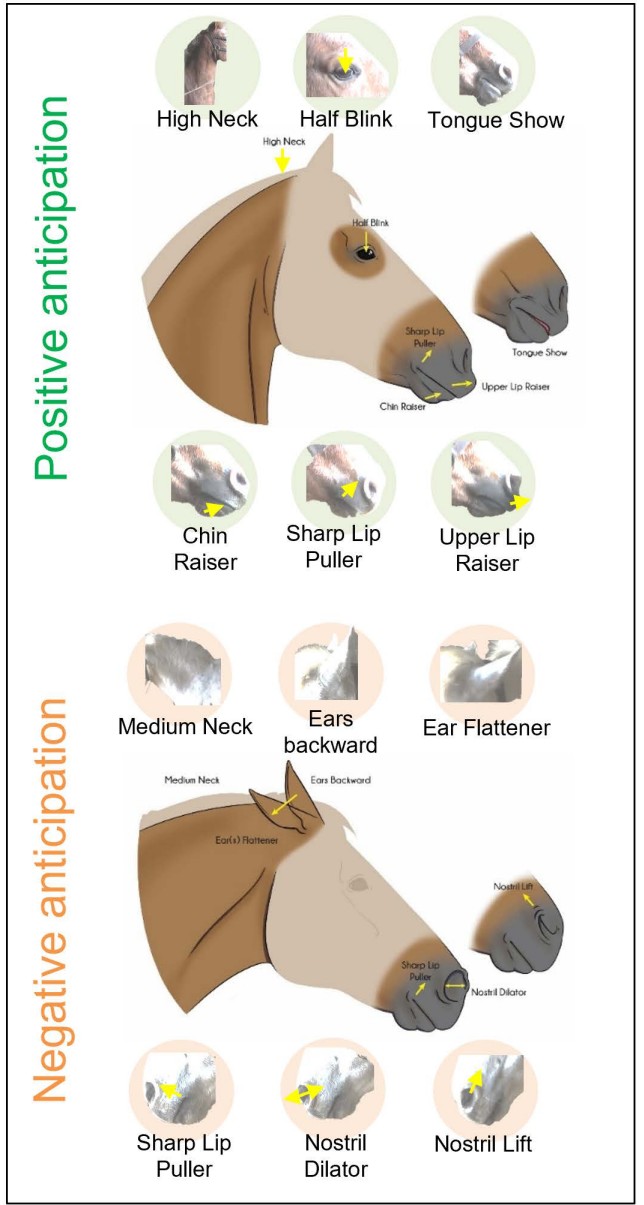

**Fig 5. Facial expression profiles related to positive and negative anticipation, composed of action units identified in 20 horses as specific to each anticipation (** Table 5**).** Non-specific facial features are shown in shaded areas in the illustration.

Nevertheless, our results revealed two action units more frequently expressed in positive anticipation that had not been previously identified in this context. The first is the 'Half-blink' (AU47), which differs from the full 'Blink' (AU145), typically linked to negative situations [37,44,80]. Unlike the full blink, the half-blink may represent a more affiliative gesture, as seen in other species like cats [81]. The second action unit that had never been associated with positive anticipation prior to this study is the 'Sharp Lip Puller' (AU113). However, an equivalent called 'lip tension' was associated with a negative context in another study [6].

The analyses conducted with the models examining the action units separately have permitted the identification of certain action units as being more expressed in one condition than another in the present study. However, given the divergent

results concerning action units found in the scientific literature, it is important when studying facial expression to analyse the combination of facial action units with each other, which provides a second level of interpretation.

We subsequently conducted a network analysis using the NetFACS to examine the action units as a set of elements that can co-occur. This approach enabled the identification of the combinations of action units with one another, as Net-FACS facilitates the analysis of data from action units within networks, rather than in isolation [51]. It is notable that action units are integral to the formation of larger facial expressions and can be combined and autocorrelated. An action unit can thus be observed in a variety of emotional contexts; however, it is the combination of action units that will define the profile of a facial expression. Consequently, two distinct networks were identified, one specific to positive anticipation and the other to negative anticipation (Fig 4).

Through the integration of the two levels of analysis (Table 5), the facial expression profile of the horse in positive anticipation was characterised by a high neck position, with the eyes making half-blinks, a sharp lip puller, upper lip and chin raiser movements, and more visible tongue. In negative anticipation, the horse exhibited a medium neck position, with the ears back, flattened more, as well as making more nostril movements by dilating or lifting them, and also making sharp lip puller movements (Fig 5).

Moreover, it was observed that some of the action units were shared by the two networks (Fig 4). However, the manner in which they were connected provided a significant amount of additional information. It is essential to consider that, even when an action unit is shared between two networks, it is the connections between them that are important. For example, in humans, the 'Inner Brow Raiser' (AU1) can be found both in the facial expression profile of surprise and in that of sadness, but it is its connections with the other action units that differ [51].

For example, 'High Neck Position' was specific to both networks. However, in negative anticipation, this neck position was associated with 'Ears forward', 'Dilated Nostrils' (AU38) and 'Nostril Lift' (AUH13). This combination of high neck and forward ears has been identified in the vigilance posture [82] (Fig 4), and the dilation of the nostrils has been observed in a context of a stressful event [80], as well as the nostril lift seen in frustration in horses [44]. In contrast, in positive anticipation, the 'High Neck Position' was linked to 'Upper Lid Raiser' (AU10) and 'Eye White Increase' (AD1), which may suggest excitement or increased attention (Fig 4). Although both types of anticipation included the high neck position, the interpretation of this feature may differ due to its connections with other action units, allowing for a divergent interpretation.

This was also the case for 'Chewing' (AD81). In the positive anticipation network, 'Chewing' (AD81) was associated with 'Tongue Show' (AD19), itself associated with 'Ears forward'. This suggests the development of a conditional response in the horses to the expectation of access to the pasture (Fig 4). As demonstrated in Pavlov's experiments with dogs, in which salivation was elicited by the conditioned stimulus of a bell sound [83], the horses in our study exhibited chewing and tongue-out behaviours in positive anticipation. It remains to be determined whether this action unit is specific to the anticipation of food (since one aspect of going to the pasture is eating grass), or whether it reflects anticipation of a positive event in general, whether it involves food or not (another aspect of going to pasture is gaining space and freedom and movement).

In contrast, 'Chewing' (AD81) was connected only with 'Ears backward' in the negative anticipation network. These two action units have been described in negative situations such as food frustration [37,44] pain [41] and negative interactions [84,85]. In this combination, chewing could therefore reflect stressful behaviour (Fig 4). Network analysis is therefore a new and innovative way of evaluating facial expressions. It allows us to observe the combinations of action units in relation to each other, providing valuable additional information that enhances and completes our findings.

## Conclusions

This study enabled the characterisation of two contrasting facial expression profiles specific to positive and negative anticipation through an experimental protocol, where the opposite emotional valence of the conditions was validated

using existing indicators. In addition to the conventional statistical analysis, NetFACS analysis, which represents an innovative approach to the analysis of facial expressions, enabled the acquisition of more detailed information about the combinations of action units, thereby facilitating a more comprehensive understanding of the activation of action units in relation to one another. Thus, the positive anticipation face was characterised by a high neck position, half-blinks, sharp lip puller, upper lip and chin raiser movements, and more visible tongue. The negative anticipation face was characterised by a medium neck position, the ears back, flattened more often, as well as more nostril movements by dilating or lifting them and sharp lip puller movements. Further studies are needed to determine if these facial expressions are very specific to the conditions used in this experiment or could be transposed to other positive and negative anticipation situations. This also raises the question of the function of these identified facial expression profiles. It seems plausible that they may serve a communicative function in interactions with both conspecifics and humans, a hypothesis that would benefit from further investigation. In conclusion, facial movements can offer insight into the emotional valence experienced by an animal, providing a window into its internal emotional processes. The ability to discern both positive and negative emotional states is crucial for the assessment of equine welfare. Facial expressions offer a valuable means of identifying positive experiences, and their incorporation can facilitate the improvement of practices aimed at enhancing the welfare of horses.

## Supporting information

**S1 Table. Results of inter-coder reliability (Intra-class correlation coefficient test and Cohen's Kappa test).**
(PDF)

**S2 Table. Results of intra-coder reliability (Intra-class correlation coefficient test and Cohen's Kappa test).**
(PDF)

## Acknowledgments

We would like to thank the staff of the UEPAO experimental unit (Unité Expérimentale de Physiologie Animale de l'Orfrasière) for their help in setting up the experimental protocol. We would also like to thank Anne-Lyse Lainé for the cortisol analyses of the saliva samples and Alexandrine Wagner for the illustrations.

## Author contributions

**Conceptualization:** Romane Phelipon, Léa Bertrand, Plotine Jardat, Fabrice Reigner, Léa Lansade.

**Data curation:** Romane Phelipon, Léa Bertrand, Plotine Jardat, Fabrice Reigner.

**Formal analysis:** Romane Phelipon, Léa Bertrand.

**Funding acquisition:** Romane Phelipon, Léa Lansade.

**Investigation:** Romane Phelipon.

**Methodology:** Romane Phelipon, Léa Bertrand, Plotine Jardat, Kate Lewis, Jérôme Micheletta.

**Project administration:** Léa Lansade.

**Supervision:** Léa Lansade.

**Validation:** Jérôme Micheletta, Léa Lansade.

**Visualization:** Kate Lewis, Léa Lansade.

**Writing – original draft:** Romane Phelipon, Léa Bertrand, Plotine Jardat, Léa Lansade.

**Writing – review & editing:** Romane Phelipon, Plotine Jardat, Kate Lewis, Jérôme Micheletta, Léa Lansade.

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
