## [Decision Letter · Decision Letter 0]

3 Mar 2025

PONE-D-25-05198Characterisation of facial expressions and behaviours of horses in response to positive and negative emotional anticipation using network analysisPLOS ONE

Dear Dr. Phelipon,

Thank you for submitting your manuscript to PLOS ONE. After careful consideration, we feel that it has merit but does not fully meet PLOS ONE’s publication criteria as it currently stands. Therefore, we invite you to submit a revised version of the manuscript that addresses the points raised during the review process.

We look forward to receiving your revised manuscript.

Kind regards,

Laura Patterson Rosa, M.V., Ph.D.

Academic Editor

PLOS ONE

Journal Requirements:

Reviewers' comments:

Reviewer's Responses to Questions

**Comments to the Author**

1. Is the manuscript technically sound, and do the data support the conclusions?

Reviewer #1: Yes

Reviewer #2: Yes

2. Has the statistical analysis been performed appropriately and rigorously? 

Reviewer #1: Yes

Reviewer #2: Yes

3. Have the authors made all data underlying the findings in their manuscript fully available?

Reviewer #1: Yes

Reviewer #2: Yes

4. Is the manuscript presented in an intelligible fashion and written in standard English?

Reviewer #1: Yes

Reviewer #2: Yes

5. Review Comments to the Author

Reviewer #1: This is an interesting study looking to assess indicators of positive and negative anticipation in horses. The study is well written and very clear to follow. The idea of regarding multiple behavioural signals at the same time is highly applicable as most horse handlers already do this (often unconsciously). Most of my comments are editorial or to provide more detail in the methodology.

L131 – can you include some justification of why going to the pasture would be perceived as positive?

L191 and L202 – it is helpful if figure captions contain all information necessary to understand the figure without the reader having to consult the text. Thus you could include information about what you are testing, what species and how many you are testing, etc. This would apply to all figures and tables in the manuscript.

L206 – can you clarify what you mean by ‘attached with a halter’ does this mean the horse was simply wearing a halter or the horse was tethered in to the halter while in the box?

L211 – ah, I see now you remove the ‘lunges’ from the halter, so it would appear the horse was tethered. Better wording would be ‘…the experimenter untethered the horse to all them….’

L214 – can you provide a reason why the experimenter was wearing a green coat? Same on L222 – why a blue coat?

L216 – clarify what you mean by ‘the device’. Also L223

L218 - clarify what you mean by ‘guided’. Was the horse led with a lead rope or by the halter?

L219 – how much concentrate was the horse given and was it a feedstuff they were familiar with?

L220 – the isolation box would only be unknown to the horses the first time they experienced it. After that it would be a familiar space.

L227 – what types of objects and how many were in the isolation box? Did they change with each time the horse entered the box? What was the decibel level of the sound recording?

L275 – what was the criteria to determine an artifact

L282 – please include some details on how the saliva samples were collected

L287 – clarify that it was the supernatant that was place din the 1ml tubes

L290 – please include information about the immunoassay (supplier, manufacturer). Also did you average the cortisol concentrations from the two 50ul aliquots for each sample?

L316 – is there a reference for the EquiFACS manual?

In Table 1 under Supplementary Behaviours, the first row is observe congener/experiment. Perhaps conspecific is a better word than congener?

L427 – here and for all results, it would be appropriate to include the actual statistical values (e.g. F-statistic) along with the p values in the written results. However since you have all the statistical values in Table 2, no need to include them in the text at all.

L476 – here you report on medium neck position, but L409 indicated that you were only analyzing high and low neck positions.

L550 – change the word ‘equine’ to ‘equids’

L577 – the question of valence vs arousal is a really important point!

L587 – neck position could also be influenced by what the horse is looking at, specifically if it is something up close/far away or above or below their line of vision. Horses must raise their head to be able to focus on an object.

L611 – please include a reference for the sentence on the purpose of NetFACS

L659 – I would leave it to the discretion of the editor but generally figures and tables should not appear in the discussion. Figure 5 might instead be very useful as a graphical abstract.

A careful review of grammar would be helpful. In some cases there is a switch between present and past tense, or word order or choice makes meanings unclear.

Check references carefully. Some are lacking some information (eg. Ref 17, 18, 46, 59, 61, 67). Many are missing DOIs

Reviewer #2: I wish to complement the authors on a well written and organized manuscript. This study is of interest and importance to the field of equine behavior and welfare and provides a novel approach in combination with previously tested/recommended behavioral and physiological indicators to gain further insight into equine affective state. Overall, the methods and results are presented clearly and provide useful information which will aid other researchers in designing studies to further investigate equine emotion for the purpose of assessing welfare and assuring horses are experiencing positive welfare in different management schemes and handling situations.

Specific recommended edits:

Line 122 - Change "methods" to "method"

In the Materials and Methods section of the manuscript, horses were randomly assigned to groups (i.e., randomly assigned to order in which they would be tested in the positive and negative conditions), correct? It would be helpful to the reader if some additional detail concerning the study horses' routine/previous access to turnout and whether a companion horse is or was typically present during turnout is included.

Lines 206-210 - The term "lunges" is not familiar in this context. Do you mean lead line? Specifically, it isn't completely clear in this statement whether horses are free-standing in the stall or tied in the stall with a halter and lead line/lead rope.

Lines 216 and 218 and later in Line 225 - Study subjects are described as Welsh mares, but here in the methods, the terms "he" and "him" are used when referring to an individual horse.

Line 331 - Change "defines" to "defined"

6. PLOS authors have the option to publish the peer review history of their article (what does this mean? ). If published, this will include your full peer review and any attached files.

**Do you want your identity to be public for this peer review?** For information about this choice, including consent withdrawal, please see our Privacy Policy .

Reviewer #1: No

Reviewer #2: No

---

## [Author Response · Author response to Decision Letter 1]

1 Apr 2025

We would like to express our sincere gratitude to the editor and reviewers. Their constructive and highly relevant comments on the manuscript have significantly contributed to its improvement and enhanced the quality of our work.

All suggested corrections have been made and are detailed in the Response to Reviewers document, with references to the corresponding line numbers in the document ‘Revised Manuscript with Track Changes’, in Revision mode: All Markup and Show Markup / Balloons / Show All Revisions Inline.

---

## [Decision Letter · Decision Letter 1]

8 Apr 2025

Characterisation of facial expressions and behaviours of horses in response to positive and negative emotional anticipation using network analysis

PONE-D-25-05198R1

Dear Dr. Phelipon,

We’re pleased to inform you that your manuscript has been judged scientifically suitable for publication and will be formally accepted for publication once it meets all outstanding technical requirements.

Kind regards,

Laura Patterson Rosa, M.V., Ph.D.

Academic Editor

PLOS ONE

Additional Editor Comments (optional):

Please address the following minor comment from the reviewers:

L169 – you mention the study took place from end of march to early February – was this a total of 11 months then? Perhaps include the years. Or perhaps the months are backward (early February to end of march)?

Reviewers' comments:

Reviewer's Responses to Questions

**Comments to the Author**

1. If the authors have adequately addressed your comments raised in a previous round of review and you feel that this manuscript is now acceptable for publication, you may indicate that here to bypass the “Comments to the Author” section, enter your conflict of interest statement in the “Confidential to Editor” section, and submit your "Accept" recommendation.

Reviewer #1: All comments have been addressed

Reviewer #2: All comments have been addressed

2. Is the manuscript technically sound, and do the data support the conclusions?

Reviewer #1: Yes

Reviewer #2: Yes

3. Has the statistical analysis been performed appropriately and rigorously? 

Reviewer #1: Yes

Reviewer #2: Yes

4. Have the authors made all data underlying the findings in their manuscript fully available?

Reviewer #1: Yes

Reviewer #2: Yes

5. Is the manuscript presented in an intelligible fashion and written in standard English?

Reviewer #1: Yes

Reviewer #2: Yes

6. Review Comments to the Author

Reviewer #1: Thank you for thoroughly addressing all comments. The only comment I had is on L169 – you mention the study took place from end of march to early February – was this a total of 11 months then? Perhaps include the years. Or perhaps the months are backward (early February to end of march)?

Reviewer #2: (No Response)

7. PLOS authors have the option to publish the peer review history of their article (what does this mean? ). If published, this will include your full peer review and any attached files.

**Do you want your identity to be public for this peer review?** For information about this choice, including consent withdrawal, please see our Privacy Policy .

Reviewer #1: No

Reviewer #2: No

---

## [Editor Report · Acceptance letter]

PONE-D-25-05198R1

PLOS ONE

Dear Dr. Phelipon,

I'm pleased to inform you that your manuscript has been deemed suitable for publication in PLOS ONE. Congratulations! Your manuscript is now being handed over to our production team.

Kind regards,

on behalf of

Dr. Laura Patterson Rosa

Academic Editor

PLOS ONE